# Work engagement and its association with emotional intelligence and demographic characteristics among nurses in Palestinian neonatal intensive care units

Ibrahim Aqtam [1]*, Ahmad Ayed [2], Ahmad Batran[3], Moath Abu Ejheisheh[3], Riham H. Melhem[3], Mustafa Shouli[1]

1 Department of Nursing, Ibn Sina College for Health Professions, Nablus University for Vocational and Technical Education, Nablus, Palestine, 2 Faculty of Nursing, Arab American University, Jenin, Palestine, 3 Faculty of Allied Medical Sciences, Department of Nursing, Palestine Ahliya University, Bethlehem, Palestine

* ibrahim.aqtam@nu-vte.edu.ps, info@nu-vte.edu.ps

## Abstract

### Introduction

Work engagement, defined as a positive, fulfilling, work-related state of mind characterized by vigor, dedication, and absorption, is crucial for nurse retention and quality of care in high-stress environments. Neonatal Intensive Care Units (NICUs) present unique emotional and psychological challenges for nurses, necessitating skills like emotional intelligence (EI) to enhance work engagement. This study investigates the association between EI, demographic factors, and work engagement among Palestinian NICU nurses.

### Methods

A cross-sectional, descriptive correlational design was employed during February-April 2025. Of 230 nurses invited, 207 completed the survey (response rate = 90.2%) across 12 Palestinian NICUs using convenience sampling. Data analysis was conducted using descriptive statistics, Pearson's correlation, and multiple linear regression via SPSS v26. Validated tools, the Schutte Self-Report Emotional Intelligence Test (SSEIT) and Utrecht Work Engagement Scale (UWES), were used.

### Results

Emotional intelligence (EI) demonstrated a strong positive correlation with work engagement ($r = 0.693$, $p < 0.001$), accounting for 48.0% of the variance in engagement scores. Age ($B = 0.463$, $\beta = 0.535$, $p = 0.002$), female gender ($B = -2.250$, $\beta = -0.115$, $p = 0.017$), and rotating shifts ($B = 1.579$, $\beta = 0.105$, $p = 0.028$) were significant predictors. EI was the strongest predictor ($B = 0.358$, $\beta = 0.593$, $p < 0.001$). The EI subdimension "utilizing emotions" scored highest ($M = 47.3 \pm 5.8$).

**Data availability statement:** The datasets used and analyzed during the current study are available from the corresponding author upon reasonable request, subject to ethical approval and institutional guidelines. To ensure long-term data stability and accessibility, requests may also be directed to the Institutional Review Board (IRB) of Palestine Ahliya University at the following contact: • Email: B.alassoud@paluniv.edu.ps (Dr. Bahaa Alassoud, IRB Chair) • Phone: +970 (2) 275 1566 (University switchboard; request IRB office) • Institutional Address: College of Allied Medical Sciences, Palestine Ahliya University, Al Daher Mountain – Bethlehem, PO: 1041. Data will be archived for a minimum of 10 years in accordance with institutional policies. External researchers must submit a formal request outlining the purpose of data use, methodology, and ethical compliance. Approval will be granted by the IRB based on alignment with the original study's ethical framework.

**Funding:** The author(s) received no specific funding for this work.

**Competing interests:** The authors have declared that no competing interests exist.

## Discussion

The findings demonstrate strong associations between EI and engagement in high-stress NICU environments. Based on these findings, we propose implementing comprehensive EI training programs in nursing curricula, establishing mentorship programs to address age-related disparities, and developing gender-sensitive work-place policies to optimize work engagement and improve patient care quality.

---

## Introduction

Work engagement among nurses in high-stress environments represents a critical challenge in healthcare delivery, particularly in specialized units requiring intense emotional labor and clinical expertise. Work engagement, defined as a positive, fulfilling, work-related state of mind characterized by vigor, dedication, and absorption [1], is vital for nurse retention and care quality. Neonatal Intensive Care Units (NICUs) are high-stress environments characterized by emotionally and psychologically demanding workloads. The research problem centers on understanding how emotional intelligence (EI) and demographic factors are associated with work engagement among Palestinian NICU nurses, given the unique challenges posed by resource constraints, political instability, and cultural factors in this context.

The magnitude of this issue is substantial, as work engagement directly impacts patient outcomes, nurse retention, and healthcare quality. Studies indicate that engaged nurses demonstrate 12% better patient outcomes and 40% lower turnover rates compared to their disengaged counterparts [1]. Nurses in these settings routinely confront challenges such as managing critically ill infants, supporting distraught families, and navigating ethical dilemmas [2]. Moreover, these challenges often intersect with broader ethical demands in clinical care, where competencies like bioethical decision-making become essential for sustaining compassionate, patient-centered practice [3]. The necessity of conducting this study arises from the gap in literature specifically addressing EI-engagement relationships in resource-constrained NICU settings, particularly in conflict-affected regions like Palestine.

The theoretical framework underlying this study is grounded in Self-Determination Theory (SDT), which posits that work engagement emerges from the satisfaction of three basic psychological needs: autonomy, competence, and relatedness [1]. Within this framework, emotional intelligence (EI) serves as a facilitating factor that enhances nurses' ability to meet these needs by improving emotional regulation, interpersonal relationships, and adaptive coping strategies in high-stress environments.

The practical purpose of this study is to provide evidence-based recommendations for healthcare administrators and policymakers to enhance nurse engagement through targeted interventions, ultimately improving both staff well-being and neonatal care quality. These stressors necessitate advanced coping skills, among which EI, the ability to perceive, regulate, and utilize emotions, has emerged as a critical competency for sustaining professional performance and well-being [4,5].

Based on the theoretical framework and existing literature, we formulated the following hypotheses: H1: EI will be positively associated with work engagement among Palestinian NICU nurses. H2: Age will be positively associated with work engagement levels. H3: Demographic characteristics (gender, education level, shift patterns) will be significantly associated with work engagement.

Empirical evidence underscores the role of EI in healthcare settings. For instance, a study by Ayed found a significant positive correlation (r = 0.53, p < 0.001) between EI and clinical decision-making accuracy among NICU nurses, highlighting its relevance to patient care [6]. Similarly, Barr demonstrated that EI subdimensions, such as mood regulation, predict reduced psychological distress (β = −0.29) and enhanced emotional well-being (β = 0.41) in this population [7]. These findings align with broader meta-analytic evidence: a recent synthesis by Lee et al. [8] confirmed that EI reduces burnout risk by 22% across nursing populations (d = 0.47, 95% CI [0.33, 0.61]), primarily through its capacity to enhance engagement and adaptive coping. Such results reinforce the centrality of EI in high-acuity nursing roles [9,10].

The theoretical framework linking EI to work engagement suggests that nurses with higher emotional intelligence can better manage workplace stressors, maintain positive relationships, and sustain motivation despite challenging conditions. In NICUs, where emotional labor is intensive, engagement may buffer against burnout and turnover. Studies suggest that EI enhances engagement by enabling nurses to manage stressors effectively and maintain positive interpersonal relationships [11,12]. For example, Gao et al. [9] reported that EI mediated 35.6% of the effect of perceived organizational support on engagement among Chinese nurses. Despite this evidence, research focusing explicitly on EI and engagement in NICUs remains sparse, particularly in low-resource settings like Palestine.

The Palestinian healthcare context adds unique dimensions to this discussion. Chronic resource shortages, political instability, and cultural norms around familial involvement in care amplify stressors for NICU nurses [13]. In such high-pressure conditions, ethical preparedness and psychological readiness for emergency scenarios become particularly relevant [14]. Moreover, the COVID-19 pandemic has further intensified demands on healthcare systems globally, including in Palestine, underscoring the importance of emotional resilience and systemic support for frontline staff [15]. Yet, no prior studies have explored how EI interacts with demographic factors to influence engagement in this population. Addressing this gap is critical for developing targeted interventions to support Palestinian NICU nurses, who are pivotal to neonatal survival in a region with limited access to specialized care.

This study therefore aims to investigate the association between EI, demographic characteristics, and work engagement among nurses in Palestinian NICUs. By identifying factors associated with engagement, the findings may inform strategies to enhance workforce resilience and care quality in resource-constrained settings.

## Methods

### Study design, setting, and sampling

This study employed a cross-sectional, descriptive correlational design to assess the relationship between EI and work engagement among Palestinian NICU nurses. The study was conducted between February 15 and April 15, 2025. The sample was collected from 12 NICUs in governmental hospitals in the West Bank that operate level II and III NICUs using a convenience sampling method. These 12 NICUs cover all geographic regions of the West Bank and operate under standardized governmental protocols, ensuring coverage of diverse clinical settings while maintaining homogeneity in organizational policies. The target population included nurses working in level II and III NICUs within all governmental hospitals across the West Bank who met the study's inclusion criteria. Private hospitals were excluded from this study due to differences in resource availability, staffing patterns, and organizational policies that could introduce confounding variables and affect the homogeneity of the sample. According to the Palestinian Ministry of Health [16], governmental hospitals in Palestine operate under standardized protocols and face similar resource constraints, making them suitable for this study's focus on public healthcare settings.

These hospitals were selected because they are leading providers of neonatal care, featuring well-established NICUs that represent nurses working in large, publicly funded healthcare hospitals. According to the Palestinian Ministry of Health, governmental hospitals in Palestine operate under standardized protocols and face similar resource constraints, making them representative of the broader public healthcare system [16]. Moreover, the standardized policies and procedures across these hospitals contribute to a consistent work environment, thereby enhancing the internal validity of the study's findings.

Data were collected from 12 NICUs that collectively housed approximately 150 incubators. A convenience sampling method was used to recruit nurses from the target population. Convenience sampling was used due to the specialized nature of NICU work, the need for voluntary participation, and practical constraints related to accessing nurses during their demanding work schedules. While this approach limits generalizability, it was the most feasible method given the study context. The required sample size was determined using Raosoft software, based on an estimated population of 400 NICU nurses. With a 95% confidence level and a 5% margin of error, the minimum sample size was calculated to be 197. To account for potential non-responses or dropouts, the sample was increased to 230 nurses to ensure adequate power for statistical analysis. Ultimately, 230 nurses were invited to participate, of whom 23 declined or were ineligible, and 10 provided incomplete responses. The final sample comprised 207 nurses, resulting in a response rate of 90.2% (Fig 1).

The inclusion criteria for the study required participants to be full-time NICU nurses who provided informed consent and had at least six months of work experience in the unit. This six-month criterion was established to ensure that participants had sufficient practical knowledge, completed their orientation period, and developed familiarity with the specific responsibilities and challenges of NICU work. Nurses were excluded if they worked part-time, were on leave, or held administrative roles. Nurses who were not proficient in English were also excluded, as the questionnaires were administered in English. The exclusion of administrators was intentional, as their responsibilities are primarily managerial and not directly involved in bedside patient care.

## Measures

The first section of the questionnaire collected demographic information, including age, gender, education level, NICU experience, and work shift. The demographic variables were limited to these four key characteristics as they represent the most relevant factors identified in previous literature on work engagement among nurses. Other variables such as income level were not included due to cultural sensitivities and potential reluctance of participants to disclose financial information in the Palestinian context. Geographical location of hospitals was controlled by including only governmental hospitals with similar resource constraints and organizational structures.

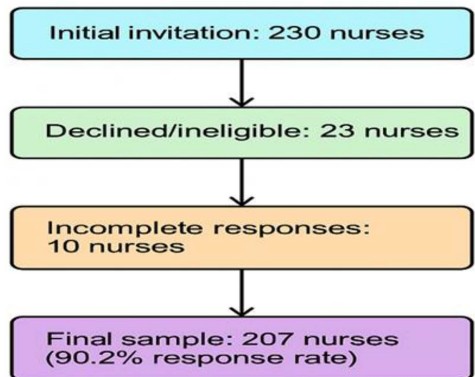

**Fig 1. Participant recruitment flowchart.**

To assess EI, the study employed the Schutte Self-Report Emotional Intelligence Test (SSEIT) [17]. Both the SSEIT and UWES were originally developed in English. Given that the questionnaires were self-administered in English, nurses who were not proficient in English were excluded from the study, as stated in the exclusion criteria. The decision to use self-administered questionnaires rather than face-to-face interviews was made to ensure standardized data collection across multiple sites, reduce interviewer bias, and allow participants to complete the survey at their convenience during their work shifts. The SSEIT was selected based on its established psychometric properties and previous validation in healthcare settings. The SSEIT consists of 33 items covering four domains: "perception of emotions," "social skills," "self-management of emotions," and "emotion utilization." Responses are rated on a five-point Likert scale, with higher scores indicating greater emotional intelligence. Total scores range from 33 to 165. The scoring system assigns points from 1 (strongly disagree) to 5 (strongly agree) for positively worded items, with reverse scoring for negatively worded items. The SSEIT has demonstrated high validity and reliability across diverse populations, with reported Cronbach's alpha values ranging from 0.84 to 0.90 [6,18–20]. In the current study, the Cronbach's alpha was 0.88.

Work engagement was measured using the 9-item Utrecht Work Engagement Scale (UWES) [1]. The UWES was chosen for its brevity, validated psychometric properties, and widespread use in nursing research. The UWES assesses three dimensions: Vigor (3 items), dedication (3 items), and absorption (3 items). Each item is rated on a seven-point scale ranging from 0 ("never") to 6 ("always"), with higher total scores indicating greater work engagement. The UWES has demonstrated satisfactory reliability and validity across multiple studies, with Cronbach's alpha values typically exceeding 0.85 [21,22]. In this study, the Cronbach's alpha for the UWES was 0.90, indicating excellent internal consistency.

Both instruments, the SSEIT for EI and the UWES for work engagement, had been previously validated in the Palestinian nursing context, ensuring cultural sensitivity and contextual appropriateness. Their prior use suggests they are reliable tools for assessing these constructs among Palestinian nurses [13,23].

## Ethical considerations and data collection procedure

Ethical approval for this study was granted by the Institutional Review Board (IRB) of Palestine Ahliya University (Ref. No# CAMS/BSN/11/2025). All data were de-identified immediately following collection and stored securely in password-protected electronic databases in compliance with international data protection standards equivalent to GDPR requirements. Data access was restricted to the research team members only. The study was conducted in accordance with the ethical principles of the Declaration of Helsinki. Prior to participation, all nurses received a detailed explanation of the study's purpose, procedures, potential benefits, and risks, and each provided written informed consent. Participation was entirely voluntary, with the option to withdraw at any point without any negative consequences. Confidentiality was strictly maintained, and all data were securely stored.

To support the data collection process, the research team visited the selected hospitals, met with the head nurses of the NICUs, explained the study's aims, and secured permission to access the units and obtain lists of eligible nurses. Time was given to complete the questionnaire in English, and sealed envelopes were provided for returning the completed forms. The researchers remained present on-site, collecting the completed forms at the end of each workday to prevent any loss or misplacement. Data collection was conducted between February 15 and April 15, 2025.

## Statistical analysis

Data were entered and analyzed using the Statistical Package for the Social Sciences (SPSS), version 26.0. Before analysis, all questionnaires were reviewed for completeness, missing data, outliers, and adherence to assumptions of normality. Normality was evaluated using both histograms and the Kolmogorov-Smirnov test. The data were found to be complete, free of outliers, and normally distributed. Prior to conducting multiple linear regression analysis, assumptions including linearity, independence of residuals, homoscedasticity, and absence of multicollinearity were assessed and confirmed.

Descriptive statistics, including means, standard deviations, frequencies, and percentages, were used to summarize the characteristics of the study variables. Independent t-tests and ANOVA were used to examine relationships between categorical variables (gender, educational level, shift pattern) and the continuous variable (work engagement). Pearson's correlation was employed to examine relationships between continuous variables only (age, experience, EI). Multiple linear regression analysis was conducted using the enter method to identify significant predictors of work engagement among NICU nurses. Correlation strengths were interpreted based on Cohen's (1988) guidelines [2]: values between 0.10–0.29 were considered weak, 0.30–0.49 moderate, and 0.50–1.0 strong. Statistical significance was determined at a p-value threshold of less than 0.05.

## Results

A total of 230 nurses were invited to participate in the study. Of these, 23 declined participation or were ineligible, and 10 provided incomplete responses, resulting in 207 nurses who completed the questionnaire (response rate = 90.2%). The average age of the participants was 32.7 ± 8.0 years. The majority (178, 86.0%) were female. Most nurses (143, 69.1%) held a bachelor's degree. Participants reported an average of 9.4 ± 7.2 years of work experience in the NICU. Additionally, the majority of participants (145, 70.0%) reported working rotating shifts (Table 1).

The overall mean score for EI was 117.6 ± 11.3, with a range of 33–165. The average EI score of 117.6 represents a moderately high level of emotional intelligence, indicating that Palestinian NICU nurses demonstrate good emotional competencies. Among the subdimensions, "utilizing emotions" had the highest mean score (47.3 ± 5.8). Work engagement showed a mean score of 34.7 ± 6.8, with scores ranging from 0–54. Within the work engagement subdimensions, dedication had the highest mean (12.5 ± 2.0) (Table 2).

The results showed that work engagement was significantly correlated with age (r = 0.264, p = 0.001), experience (r = 0.229, p = 0.001), and emotional intelligence (r = 0.693, p < 0.001). Male nurses reported higher engagement (M = 38.3, SD = 6.4) than females (M = 34.1, SD = 6.7; p = 0.002), while nurses on rotating shifts (M = 35.6, SD = 6.5) scored significantly higher than those on day shifts (M = 32.6, SD = 7.1; p = 0.003). Additionally, educational level showed a significant effect (F = 7.199, p = 0.001), with master's and above (M = 38.0, SD = 7.5) and bachelor's degree holders (M = 35.3, SD = 6.7) reporting higher engagement than diploma nurses (M = 31.8, SD = 6.1). The Tukey HSD results show that nurses with a diploma reported significantly lower work engagement compared to those with a bachelor's (p = 0.005) or master's degree (p = 0.003). However, no significant difference was found between bachelor's and master's degree holders (p > 0.05), as seen in (Table 3).

Multiple linear regression analysis was performed to examine the predictors of work engagement among nurses. Prior to analysis, all regression assumptions were tested and met, including linearity, independence of residuals,

**Table 1. Demographic characteristics of Palestinian NICU nurses (N = 207), February–April 2025.**

| Characteristics | Categories | N (%) | M (SD) |
|---|---|---|---|
| **Age** | | | 32.7 ± 8.0 |
| **Gender** | Male | 29 (14.0) | |
| | Female | 178 (86.0) | |
| **Educational level** | Diploma | 47 (22.7) | |
| | Bachelor's | 143 (69.1) | |
| | Master's and above | 17 (8.2) | |
| **Work experience in NICU** | | | 9.4 ± 7.2 |
| **Shift-work** | Day shift | 62 (30.0) | |
| | Rotating shift | 145 (70.0) | |

N: Number; %: Percentage; M: Mean; SD: Standard Deviation.

**Table 2. Emotional intelligence and work engagement scores among Palestinian NICU nurses (N = 207), February-April 2025.**

| Variable | M (SD) | Range |
|---|---|---|
| **Emotional intelligence** | 117.6 ± 11.3 | 33–165 |
| Perception of emotions | 19.1 ± 2.5 | |
| Social skills | 24.5 ± 2.7 | |
| Managing self-emotions | 26.7 ± 3.3 | |
| Utilizing emotions | 47.3 ± 5.8 | |
| **Work Engagement** | 34.7 ± 6.8 | 0–4 |
| Vigor | 11.0 ± 3.1 | |
| Dedication | 12.5 ± 2.0 | |
| Absorption | 11.3 ± 2.3 | |

M: Mean; SD: Standard Deviation.

homoscedasticity, and absence of multicollinearity (VIF < 2.0 for all variables). The model was statistically significant (p < 0.001), with an $R^2$ of 0.574 and an adjusted $R^2$ of 0.561, indicating that the predictors accounted for 57.4% of the variance in work engagement scores. The enter method was used for variable selection in the multiple linear regression analysis. The multiple linear regression analysis revealed several significant factors associated with work engagement. Age was a positive and significant predictor (B = 0.428, p = 0.004), indicating that older individuals reported higher levels of work engagement. Gender, with males as the reference group, showed a significant negative association (B = −2.230, p = .031), suggesting that males reported lower engagement compared to females. Working on a rotating shift was a positive and significant predictor (B = 1.470, p = .040), indicating that employees working rotating shifts had higher engagement levels. Emotional intelligence was the strongest predictor, showing a highly significant positive association with work engagement (B = 0.357, p < .001), as seen in Table 4.

## Discussion

This study examined the association between EI and work engagement among nurses in Palestinian Neonatal Intensive Care Units (NICUs), revealing critical insights into how psychological competencies and demographic factors are associated with engagement in high-stress healthcare environments. The findings demonstrated a strong positive correlation

**Table 3. Correlation and differences between variables and work engagement among Palestinian NICU nurses (N = 207), February–April 2025.**

| Variable | | N | M | SD | Statistical test | P. Value | 95% Confidence Interval | |
|---|---|---|---|---|---|---|---|---|
| Age | | | | | r = 0.264 | 0.001** | Lower | Upper |
| Experience | | | | | r = 0.229 | 0.001** | | |
| Emotional intelligence | | | | | r = 0.693 | 0.000** | | |
| Gender | Male | 29 | 38.3 | 6.4 | t (205) = 3.118 Cohen's d = 0.63 | 0.002** | 1.53140 | 6.79716 |
| | Female | 178 | 34.1 | 6.7 | | | | |
| Working shift | Day shift | 62 | 32.6 | 7.1 | t (205) = −3.004 Cohen's d = 0.45 | 0.003** | −5.04295 | −1.04626 |
| | Rotating shift | 145 | 35.6 | 6.5 | | | | |
| Educational level | Diploma | 47 | 31.8 | 6.1 | F (2, 204) = 7.199 η² = 0.066 | 0.001** | 30.0156 | 33.6014 |
| | Bachelor | 143 | 35.3 | 6.7 | | | 34.1986 | 36.4028 |
| | Master and above | 17 | 38.0 | 7.5 | | | 34.1610 | 41.8390 |

**p < 0.01; t = Independent t-tests, F = ANOVA test.

 

**Table 4. Factors associated with work engagement: Multiple linear regression analysis among Palestinian NICU nurses (N = 207), February–April 2025.**

| Model | B | Beta | t-test | p-value | 95.0% Confidence Interval | |
|---|---|---|---|---|---|---|
| | | | | | Lower Bound | Upper Bound |
| Age | .428 | .495 | 2.952 | .004 | .142 | .713 |
| Gender (Male) | −2.030 | −.104 | −2.177 | .031 | −3.869 | −.191 |
| Education Level | 1.236 | .095 | 1.943 | .053 | −.018 | 2.490 |
| Experience in NICU | −.282 | −.296 | −1.793 | .074 | −.593 | .028 |
| Work Shift (Rotation) | 1.470 | .098 | 2.069 | .040 | .069 | 2.872 |
| Emotional intelligence | .357 | .591 | 11.910 | .000 | .298 | .416 |

$R = 0.757$, $R^2 = 0.574$, Adjusted $R^2 = 0.561$, $F = 43.936$, $P = 0.000$

B: Unstandardized regression coefficient; β: Standardized regression coefficient; CI: Confidence Interval.

between EI and work engagement ($r = 0.693$, $p < 0.001$), with EI being strongly associated with engagement, accounting for a substantial portion of the variance in scores. This aligns with prior research in critical care settings, such as Malak et al. [11], who found that emotional intelligence was significantly correlated with work engagement among Palestinian emergency nurses during crises, and Gao et al. [12], who identified EI as a mediator between organizational support and engagement. The heightened explanatory power of EI in this study likely stems from the unique demands of NICUs, where nurses navigate parental distress, ethical dilemmas, and life-or-death decisions daily, a context requiring advanced emotional regulation [2,4].

When compared to international studies, the EI score of 117.6 found in our study represents a moderately high level of emotional intelligence among Palestinian NICU nurses. This score is comparable to findings from other critical care nursing populations internationally, suggesting that despite challenging sociopolitical conditions in Palestine, nurses maintain strong emotional competencies. However, the current sociopolitical challenges in Palestine may have contributed to enhanced resilience and emotional intelligence among Palestinian nurses, as they must navigate additional stressors related to political instability, resource constraints, and social pressures while maintaining high-quality patient care. This unique context may explain why EI shows such a strong association with work engagement in our study compared to findings from more stable healthcare environments.

Comparing our results with international literature, Al Btoush et al. found similar patterns among Jordanian critical care nurses, where emotional intelligence was significantly associated with clinical decision-making and work engagement [24]. Similarly, Sawalma et al. reported comparable findings among Palestinian critical care nurses, demonstrating that EI was associated with reduced burnout while being linked to enhanced engagement [25].

The strength of this relationship can be attributed to three mechanisms. First, emotionally intelligent nurses excel in recognizing and regulating emotions, enabling them to maintain psychological stability amid workplace stressors [7]. For instance, a nurse with high EI might reframe grief over a neonatal loss into motivation to advocate for improved protocols, sustaining engagement through adversity. Second, EI is associated with collaborative relationships with colleagues and families, creating a supportive social environment that buffers against burnout [5]. Third, EI is linked to adaptive coping strategies, such as problem-focused approaches, which help nurses balance emotional labor with task absorption [12]. These mechanisms are particularly relevant in Palestinian NICUs, where political instability and resource shortages compound workplace stressors, making emotional regulation skills even more critical for maintaining engagement [26,27].

The study's findings highlight "utilizing emotions" as the highest-scoring EI subdimension ($M = 47.3 \pm 5.8$), representing a potentially important factor for reducing burnout and enhancing engagement. This finding is particularly significant as it suggests that Palestinian NICU nurses have developed strong capabilities in channeling emotional experiences into productive workplace behaviors, which may serve as a cultural adaptation to chronic stressors. Nurses with heightened

emotional intelligence can channel intense emotions, such as anxiety or grief, into purposeful clinical actions. For example, a nurse managing a preterm infant's deteriorating condition might transform distress into hyper-vigilance, systematically monitoring vital signs while maintaining composure. This ability to harness emotions adaptively aligns with Ayed's observation that EI is associated with enhanced clinical decision-making speed and accuracy under pressure [6]. Such emotional regulation not only may improve patient outcomes but also may foster psychological resilience, potentially buffering against burnout by reframing stressors as challenges rather than threats [7,8].

Demographic predictors showed additional associations with these findings. Age was positively associated with engagement (B = 0.463, p = 0.002), potentially reflecting older nurses' accumulated resilience and mentorship roles [28,29]. However, experience did not reach statistical significance in the regression model (B = −.298, p = .061), suggesting that age-related factors such as emotional maturity and life experience, rather than mere tenure in the NICU, may be the primary drivers of engagement. Gender-related differences revealed female nurses as having lower engagement (B = −2.250, p = .017), which may reflect Palestine's gendered caregiving norms and global trends where women often shoulder greater emotional labor [30,31]. Yet, this finding also reflects systemic challenges where male nurses may face cultural barriers and role ambiguity in Palestinian healthcare settings, potentially contributing to differences in engagement levels [32].

Contrary to global studies that typically associate shift work with burnout and decreased engagement [33], rotating shifts in this study were associated with higher engagement (B = 1.579, p = 0.028). This finding warrants further investigation as several unmeasured factors may explain this surprising positive association, including differences in shift length, staffing ratios, and the voluntary nature of rotation assignments in the Palestinian context. Future research should investigate these potential mediating factors, including whether shorter shift rotations, better team cohesion during rotating shifts, or the perception of rotating shifts as providing variety and skill development contribute to this association. First, rotational schedules expose nurses to diverse clinical scenarios, potentially enhancing competence and reducing monotony, which may serve as a key engagement driver in understaffed settings [34]. Second, chronic resource shortages necessitate flexible staffing approaches, making rotational shifts a pragmatic norm that nurses may perceive as empowering rather than burdensome. Third, cultural resilience factors, including strong communal support systems and collective coping strategies, may mitigate the traditional strain associated with shift work [26]. However, consistent with World Health Organization guidelines [35], prolonged rotations without adequate recovery periods can increase fatigue-related risks, necessitating careful balance between engagement benefits and worker safety.

Our findings contribute to the growing body of literature on healthcare workforce resilience in conflict-affected regions, demonstrating that contextual factors significantly influence traditional associations between working conditions and employee outcomes. The positive relationship between rotating shifts and engagement may reflect adaptive organizational responses to chronic resource constraints, where flexibility becomes a valued organizational characteristic rather than a stressor. However, consistent with World Health Organization guidelines [36], prolonged rotations without adequate recovery periods can increase fatigue-related risks, necessitating a careful balance between engagement benefits and worker safety.

When integrating cultural factors specific to Palestinian society, several elements emerge as particularly relevant. The concept of "sumud" (steadfastness) in Palestinian culture may contribute to nurses' ability to maintain engagement despite challenging conditions. Additionally, strong family and community support networks may provide emotional resources that buffer against work-related stress, enabling nurses to maintain higher levels of engagement even in demanding shift patterns.

## Implications

Based on the associations identified in this study, several key implications emerge for nursing practice, education, and policy. First, healthcare institutions should consider prioritizing the implementation of comprehensive EI training programs,

with particular emphasis on the "utilizing emotions" component that demonstrated the highest scores among participants. These programs warrant evaluation for potential integration into both pre-licensure nursing curricula and continuing education requirements for practicing nurses.

Second, given the age-related associations with work engagement, healthcare organizations should consider establishing structured mentorship programs that pair experienced nurses with younger staff members, facilitating knowledge transfer and emotional support. Third, gender-sensitive workplace policies warrant development to address the unique challenges faced by male nurses in Palestinian healthcare settings, including role clarification and professional development opportunities.

Fourth, while rotational shift patterns showed positive associations with engagement in this context, healthcare administrators should carefully monitor workload distribution and ensure adequate recovery periods to prevent burnout. Finally, policymakers should consider the broader implications of these findings for workforce planning and resource allocation in conflict-affected healthcare systems.

For future research, several directions merit investigation. Longitudinal studies should examine how EI training interventions impact work engagement over time among NICU nurses. Cross-cultural comparative studies could explore how findings from Palestinian settings translate to other conflict-affected regions. Additionally, mixed-methods research incorporating qualitative perspectives could provide deeper insights into the mechanisms underlying the relationship between rotational shifts and engagement in resource-constrained environments.

### Study strengths and limitations

This study has several strengths, including its comprehensive coverage of all governmental NICUs in the West Bank, high response rate (90.2%), and use of validated instruments with excellent reliability in this population. The study provides the first examination of EI-work engagement relationships specifically among Palestinian NICU nurses, addressing a critical gap in the literature.

However, several limitations must be acknowledged. First, the cross-sectional design prevents causal inferences about the relationships observed. Second, convenience sampling may introduce selection bias, and the exclusion of part-time nurses and those on leave may have resulted in the exclusion of nurses with lower engagement levels, potentially affecting the representativeness of findings. Third, the exclusion of private hospitals limits generalizability to the entire Palestinian nursing workforce. Fourth, the reliance on self-reported measures may be subject to social desirability bias, particularly for EI and engagement constructs. Fifth, the study did not account for potentially important confounding variables such as organizational culture, leadership styles, or specific unit-level characteristics that may influence work engagement. Finally, the study did not assess potential biases such as self-selection bias (where nurses with higher engagement may have been more likely to participate) or non-response bias (where nurses who declined participation may differ systematically from participants).

### Conclusion

This study provides compelling evidence that emotional intelligence is strongly associated with work engagement among Palestinian NICU nurses, accounting for a substantial portion of the variance in engagement scores. Age, gender, and shift patterns also demonstrated significant associations with engagement levels. These findings highlight the critical importance of emotional competencies in sustaining nurse engagement within high-stress neonatal care environments, particularly in resource-constrained settings facing additional sociopolitical challenges.

The results suggest that interventions targeting emotional intelligence development may represent a promising approach to enhancing work engagement, potentially improving both nurse well-being and patient care quality. Healthcare organizations should consider integrating EI training into professional development programs and creating supportive

work environments that foster emotional resilience. Future research should employ longitudinal designs to establish causal pathways and explore the effectiveness of targeted interventions in diverse healthcare contexts.

## Supporting information

**S1 File. SPSS syntaxoutput.**
(ZIP)

## Author contributions

**Conceptualization:** Ibrahim Aqtam, Ahmad Ayed, Moath Abu Ejheisheh.

**Data curation:** Ahmad Batran, Riham H. Melhem.

**Formal analysis:** Ahmad Ayed, Ahmad Batran.

**Investigation:** Ibrahim Aqtam, Ahmad Ayed, Mustafa Shouli.

**Methodology:** Ibrahim Aqtam, Ahmad Ayed, Moath Abu Ejheisheh.

**Project administration:** Ibrahim Aqtam.

**Supervision:** Moath Abu Ejheisheh, Mustafa Shouli.

**Writing – original draft:** Ahmad Ayed, Riham H. Melhem.

**Writing – review & editing:** Ibrahim Aqtam, Ahmad Batran, Moath Abu Ejheisheh.

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
