## [Decision Letter · Decision Letter 0]

14 Jul 2025

PONE-D-25-23550Work Engagement and Its Association with Emotional intelligence and Demographic Characteristics among Nurses in Palestinian Neonatal Intensive Care UnitsPLOS ONE

Dear Dr. Aqtam,

Thank you for submitting your manuscript to PLOS ONE. After careful consideration, we feel that it has merit but does not fully meet PLOS ONE’s publication criteria as it currently stands. Therefore, we invite you to submit a revised version of the manuscript that addresses the points raised during the review process.

We look forward to receiving your revised manuscript.

Kind regards,

Osama Mohamed Elsayed Ramadan

Academic Editor

PLOS ONE

Journal Requirements:

Reviewers' comments:

Reviewer's Responses to Questions

**Comments to the Author**

1. Is the manuscript technically sound, and do the data support the conclusions?

Reviewer #1: Yes

Reviewer #2: Yes

2. Has the statistical analysis been performed appropriately and rigorously? 

Reviewer #1: Yes

Reviewer #2: Yes

3. Have the authors made all data underlying the findings in their manuscript fully available?

Reviewer #1: Yes

Reviewer #2: Yes

4. Is the manuscript presented in an intelligible fashion and written in standard English?

Reviewer #1: Yes

Reviewer #2: No

5. Review Comments to the Author

Reviewer #1: This is an interesting paper and the content is very good but I suggest the paper can be improved in the following ways: Abstract

-Please correct all parts of the article according to the guidelines of the journal authors guideline In the methods section please bring year of performing of this study, sampling methods and data analysis methods -In the conclusion part, it is necessary to specify the researcher's proposal to improve the conditions and use of the beneficiaries

Introduction

Please bring the following items 1- Definition of the research problem 2- The magnitude and importance of the study variable 3- Expressing the necessity of conducting the study Finally, the practical purpose of the study should be stated. Method

Please report the details scoring and validations of study tools

Discussion

In the discussion section, it is necessary y to compare the main results of the study with the results of other studies in this field. To strengthen the article, especially in the introduction and discussion section the following studies are suggesteplease used and add to this manuscript references.

- The effect of bio ethical principles education on ethical attitude of prehospital paramedic personnel

- Crowd Simulations and Determining the Critical Density Point of Emergency Situations

-Explore pre-hospital emergency challenges in the face of the COVID-19 pandemic: A quality content analysis in the Iranian context

Conclusion � What are your suggestion for future studies? Best regards

Reviewer #2: Dear Authors

After reviewing the manuscript, I have the following commnets, which could enhance the manuscript.

- Paper needs proofreading and editing.

- Abstract: Introduction: add rationale for conducting the study. Methods: modify design to a cross-sectional,

descriptive correlational instead of a cross-sectional, correlational. Add time for conducting the study.

- Introduction: The study variables are fragmented and need to be correlated with each other to explain the

framework of the study. Significance of the study needs to be clarified. You wrote the introduction as literature

review, thus, no need to write all results of the previous studies. Use the full name of emotional inteliigence (EI) for

the first time then use abbreviation in all manuscript.

- Methods: Clarify why you excluded private hospitals. Add references about description of governmental hospitals in

Palestine. You mentioned the word of representative while the sample was a conveience sample. Why you increase

sample to 230. Why you selected nurses with at least 6 months, clarify that. Replace the word instruments by

measurements. Add reliability of UWES. Checj the Cohen's guidelines values and add a reference.

- Results: there was a repetition of the statment about response rate. Check this statement "corrected from 33-65??

You should add assumptions of regression and results of these assumptions according to these assumptions check

table number 3. Only correlated vaiables enter the regression model. Add interpretation of predictors variance.

- Discussion: you need to provide more depth discussion of these variables especially in conflict areas such as Palestine

and integrate cultural factors.

- Implications: focus on the significant results. What about future studies.

- Table 3 check the regression assumptions. Add all results in details (standardized regression coefficient, t-test).

Add * for significant results according to p-value.

6. PLOS authors have the option to publish the peer review history of their article (what does this mean? ). If published, this will include your full peer review and any attached files.

**Do you want your identity to be public for this peer review?** For information about this choice, including consent withdrawal, please see our Privacy Policy .

Reviewer #1: No

Reviewer #2: No

---

## [Author Response · Author response to Decision Letter 1]

15 Jul 2025

Dear Editor and Reviewer

We sincerely thank the Editor and Reviewers for their time, thoughtful feedback, and constructive suggestions. We have revised the manuscript accordingly and provide a point-by-point response file. All changes are highlighted in the revised manuscript, and page numbers refer to the tracked-changes version.

Dr Aqtam

---

## [Decision Letter · Decision Letter 1]

11 Aug 2025

PONE-D-25-23550R1Work Engagement and Its Association with Emotional intelligence and Demographic Characteristics among Nurses in Palestinian Neonatal Intensive Care UnitsPLOS ONE

Dear Dr. Aqtam,

Thank you for submitting your manuscript to PLOS ONE. After careful consideration, we feel that it has merit but does not fully meet PLOS ONE’s publication criteria as it currently stands. Therefore, we invite you to submit a revised version of the manuscript that addresses the points raised during the review process.

We look forward to receiving your revised manuscript.

Kind regards,

Osama Mohamed Elsayed Ramadan

Academic Editor

PLOS ONE

Journal Requirements:

**Additional Editor Comments:**

Thank you for submitting your revised manuscript (“Work Engagement and Its Association with Emotional Intelligence and Demographic Characteristics among Nurses in Palestinian Neonatal Intensive Care Units”) to PLOS ONE. Your dedication to exploring emotional intelligence (EI) in such a high-stakes environment is evident, and the study addresses an important gap. However, before a final decision can be reached, several substantive issues must be addressed. I invite you to submit a major revision that responds point-by-point to the comments below.

1. The Abstract states data were collected from 207 nurses, but the Methods note that 230 were invited with a 90.2% response rate. Please ensure consistency:

2. Update the Abstract to reflect the invitation and response process (e.g., “Of 230 nurses invited, 207 completed the survey [response rate = 90.2%]”).

3. Confirm the number of exclusions (initial invitation = 230; declines/ineligible = 23; incomplete = 10; final N = 207) and align these figures throughout.

4. Convenience sampling in a conflict-affected region may introduce bias. Discuss limitations more explicitly:

5. How might voluntary participation and exclusion of private hospitals affect representativeness?

6. What steps did you take to mitigate selection bias?

7. Temper any causal language (“predictors of engagement”) given the cross-sectional design.

8. In the multiple regression, experience (B = –0.298, p = .061) lost significance. Yet it remains discussed as though meaningful. Revise the narrative to reflect its non-significance.

9. For all predictors, please report standardized coefficients (β) alongside unstandardized (B) for clarity.

10. In Tables 3–4, ensure p-values are consistently formatted (e.g., p < .001 rather than p = .000).

11. The Introduction outlines EI and engagement well, but the theoretical linkage (e.g., Self-Determination Theory) is implicit. Strengthen this by:

12. Explicitly naming and citing the theoretical model underpinning your hypotheses.

13. Formulating clear hypotheses tied to that framework (e.g., H1: EI will positively correlate with work engagement; H2: Age will moderate this relationship).

14. While your findings align with previous studies, the Discussion sometimes reiterates results rather than critically engaging with them. Please revise to:

15. Delineate what is truly novel about the Palestinian NICU context versus other critical care settings.

16. Offer concrete, evidence-based recommendations for EI training (e.g., curriculum content, delivery mode).

17. Address the surprising positive association with rotational shifts: Could unmeasured factors (e.g., shift length, staffing ratios) explain this? Suggest avenues for future research.

18. You state IRB approval (Ref. CAMS/BSN/11/2025) and consent procedures; please add a brief sentence to the Ethical Statement confirming data de-identification and storage protocols in compliance with GDPR or equivalent data-protection standards.

19. Add “response rate” and specify the cross-sectional, descriptive correlational design in Keywords (e.g., “cross-sectional,” “convenience sampling”).

20. Use either “rotational shift” or “rotating shift” uniformly.

21. Standardize the naming of EI subdimensions (e.g., “utilization of emotions” vs. “utilizing emotions”).

22. Add a flowchart (as Figure 1) illustrating participant recruitment and exclusions for transparency.

23. A few sentences are lengthy and may obscure meaning. For example, split compound-complex sentences in the Introduction (L45–L50) into two for readability.

24. Check for minor typos (e.g., “gender-sensitive” appears once without a hyphen).

Decision: Major Revision

Your work holds promise for both academic and practical audiences. Please address each point above in a detailed response letter, indicating line-by-line changes in the manuscript. I look forward to reviewing your comprehensive revision.

Reviewers' comments:

Reviewer's Responses to Questions

**Comments to the Author**

1. If the authors have adequately addressed your comments raised in a previous round of review and you feel that this manuscript is now acceptable for publication, you may indicate that here to bypass the “Comments to the Author” section, enter your conflict of interest statement in the “Confidential to Editor” section, and submit your "Accept" recommendation.

Reviewer #3: All comments have been addressed

Reviewer #4: All comments have been addressed

Reviewer #5: (No Response)

2. Is the manuscript technically sound, and do the data support the conclusions?

Reviewer #3: Yes

Reviewer #4: Yes

Reviewer #5: Partly

3. Has the statistical analysis been performed appropriately and rigorously? 

Reviewer #3: Yes

Reviewer #4: Yes

Reviewer #5: Yes

4. Have the authors made all data underlying the findings in their manuscript fully available?

Reviewer #3: No

Reviewer #4: Yes

Reviewer #5: No

5. Is the manuscript presented in an intelligible fashion and written in standard English?

Reviewer #3: Yes

Reviewer #4: Yes

Reviewer #5: Yes

6. Review Comments to the Author

Reviewer #3: 1) Methods

Sampling:

i. Does the sample collection from 12 NICUs represent and cover all government hospitals in the West Bank?

ii. Justify why convenience sampling was used instead of random sampling. What were the sampling methods used in selecting the NICUs and the nurses?

Instruments & Data Collection:

i. Were both the SSEIT and UWES originally in English? If so, were they translated into Arabic for this study? What about nurses who were not proficient in English—were they excluded? If yes, this should be clearly stated in the exclusion criteria.

ii. Justify why the questionnaire was self-administered instead of using a face-to-face interview, especially to address potential technical or language issues.

2) Results

i. The title of Table 1 is incomplete. Please write the full title, including the study population, location, and study period.

ii. Justify why the demographic variables were limited to only four. What about other potentially relevant variables, such as income level or the geographical location of the hospital?

iii. The titles of Tables 2 and 3 are also incomplete. Please revise them to follow the same format as Table 1.

iv. There is a critical error in Table 3. Pearson correlation cannot be used for categorical independent variables such as gender or educational level. Both the independent and dependent variables must be continuous for Pearson correlation to be valid. In this case, t-tests or ANOVA would be more appropriate.

v. For the linear regression, please state which variable selection method was used—stepwise, backward, or forward selection? Also, the significant predictor (EI) is not mentioned in the abstract’s results section and should be included.

3) Discussion

i. Do not repeat statistical results in the discussion section. Instead, compare your findings with those of previous studies.

ii. The average EI score in your study was 117.6. Is this considered high or indicative of good emotional intelligence? How does this score compare to those reported among nurses in other countries? Could the current sociopolitical challenges in Palestine affect the EI of Palestinian nurses compared to nurses in other settings? Please discuss.

iii. What are the strengths and limitations of your study? These should be presented before the conclusion.

4) Conclusion

i. The conclusion lacks a clear summary of the study findings. It should directly answer the research objectives. Did demographic factors and emotional intelligence show an association with work engagement among nurses? Please clearly state the predictors identified.

Reviewer #4: The manuscript presents a well-structured and clearly written study that addresses an important and relevant topic in the field. The research question is well-defined, and the methodology is appropriate and adequately described. The results are presented clearly with well-organized tables and figures, and the discussion appropriately links the findings to existing literature.

Overall, the paper demonstrates scientific rigor and offers valuable contributions to the field. The writing is generally clear and concise, and the logical flow of sections makes it easy to follow.

Reviewer #5: Thank you for the opportunity to read and review this study on emotional intelligence and work engagement of NICU nurses in Palestine. I think it is well-designed and well-written, and adds important scientific knowledge.

However, I do think that some revisions must be made to the paper prior to publishing.

Major revisions:

1. The authors state correctly in the paper that their data shows association and correlation between emotional intelligence and work engagement, as well as other factors. However, the implications section and the conclusions are phrased as though the study shows a causal relationship, but it doesn't. Therefore, a sentence like "this study provides compelling evidence that emotional intelligence [...] serves as a critical protective factor in high-stress healthcare environment" is simply not supported by the results presented in the study. I suggest to rephrase these two sections to a language that discusses association rather than causation. The same is true in the discussion section in the abstract.

2. The discussion section is lacking a limitation paragraph which is customary in any published clinical study. This study has limitations, and they should be mentioned and discussed in the paper. For example, the fact that nurses who work part time or on leave were excluded might cause a selection bias, because maybe the nurses who have the least work engagement are included in this group, and this might affect the results.

Minor revisions:

1. I recommend to delete the first sentence of the abstract starting with "the rationale for the study..." as I am not sure it adds to the introduction much.

2. It would be beneficial to define work engagement right at the beginning of the introduction, rather than later in the paper.

3. Shift patterns are not a demographic characteristic, so I would suggest to remove them from the brackets in the penultimate paragraph of the introduction.

4. In the methods, the author mention inclusion criteria for participating hospitals but don't detail what they are. I suggest detailing what these criteria are, or just mention that only governmental hospitals were chosen for participation.

5. The sentence on the exclusion of private hospitals should be moved to the end of the paragraph, as it cuts the flow of the explanation about the included hospitals.

6. There is a grammatical error in the paragraph below table 3. It should be "analyses were" instead of "analysis was" (because the use of the word "multiple").

7. PLOS authors have the option to publish the peer review history of their article (what does this mean? ). If published, this will include your full peer review and any attached files.

**Do you want your identity to be public for this peer review?** For information about this choice, including consent withdrawal, please see our Privacy Policy .

Reviewer #3: No

Reviewer #4: **Yes: ** Mohammed Musaed Aljabri

Reviewer #5: No

---

## [Author Response · Author response to Decision Letter 2]

12 Aug 2025

Thank you for your thorough review and constructive feedback. Please see the attached point-by-point response addressing all editor and reviewer comments.

Dr Aqtam

---

## [Decision Letter · Decision Letter 2]

3 Sep 2025

PONE-D-25-23550R2Work Engagement and Its Association with Emotional intelligence and Demographic Characteristics among Nurses in Palestinian Neonatal Intensive Care UnitsPLOS ONE

Dear Dr. Aqtam,

Thank you for submitting your manuscript to PLOS ONE. After careful consideration, we feel that it has merit but does not fully meet PLOS ONE’s publication criteria as it currently stands. Therefore, we invite you to submit a revised version of the manuscript that addresses the points raised during the review process.

We look forward to receiving your revised manuscript.

Kind regards,

Osama Mohamed Elsayed Ramadan

Academic Editor

PLOS ONE

Journal Requirements:

Additional Editor Comments:

Decision: Revise (minor). Please address items below.

1) Bivariate analyses and Table 3 (Reviewer 3)

Re-analyze work engagement against categorical predictors using the correct tests:

Gender, shift: independent t tests. Report n, mean ± SD for each group, t, df, exact p, and Cohen’s d (with 95% CI if available).

Educational level: one-way ANOVA (use Welch if variances are unequal). Report F, df, exact p, and η² or ω²; add appropriate post-hoc comparisons if overall p < 0.05.

Keep Pearson correlations only for continuous–continuous pairs (age, experience, EI with engagement).

Replace Table 3 with the corrected outputs and adjust the accompanying Results text (one short paragraph) to match the new statistics.

Transparency: Upload your SPSS syntax/output (e.g., .sps/. spv) as a Supplement.

Note: Your regression section can stay as is (assumptions checked; VIF < 2; R²/adjusted R² and CIs reported). Only the bivariate section/table needs correction.

2) Conclusion language (Reviewer 5)

Rewrite the Conclusion to avoid any causal framing in this cross-sectional study. Replace phrases like “protective factor,” “compelling evidence,” and “emerges as a cornerstone” with association-focused wording.

Example acceptable phrasing:

“Emotional intelligence was strongly associated with higher work engagement among NICU nurses.”

“These cross-sectional findings suggest EI may be an important correlate; longitudinal or experimental studies are needed to test causal effects.”

“Programs to enhance EI warrant evaluation for potential impact on engagement and patient outcomes.”

Also, these minor issues:

1. Abstract: Specify the design precisely (“cross-sectional, descriptive correlational”) and include data-collection dates.

2. Terminology: First use “emotional intelligence (EI)”, then EI consistently throughout.

3. Methods wording: Do not call a convenience sample “representative.” Justify excluding private hospitals and cite a Ministry/official source describing governmental hospitals; state why invitations were increased to 230; justify the ≥6-month experience criterion; rename “Instruments” to “Measures”; include UWES reliability here; cite Cohen’s guidelines for effect-size interpretation.

4. Results housekeeping: Remove any duplicated response-rate sentence; in regression reporting, include standardized β, t, 95% CIs, exact p, and mark significance with * in the table notes; briefly interpret variance explained (R²/adj-R²) in plain language.

Reviewers' comments:

Reviewer's Responses to Questions

**Comments to the Author**

1. If the authors have adequately addressed your comments raised in a previous round of review and you feel that this manuscript is now acceptable for publication, you may indicate that here to bypass the “Comments to the Author” section, enter your conflict of interest statement in the “Confidential to Editor” section, and submit your "Accept" recommendation.

Reviewer #3: (No Response)

Reviewer #4: All comments have been addressed

Reviewer #6: (No Response)

2. Is the manuscript technically sound, and do the data support the conclusions?

Reviewer #3: Yes

Reviewer #4: Yes

Reviewer #6: Yes

3. Has the statistical analysis been performed appropriately and rigorously? 

Reviewer #3: No

Reviewer #4: Yes

Reviewer #6: Yes

4. Have the authors made all data underlying the findings in their manuscript fully available?

Reviewer #3: No

Reviewer #4: Yes

Reviewer #6: Yes

5. Is the manuscript presented in an intelligible fashion and written in standard English?

Reviewer #3: Yes

Reviewer #4: Yes

Reviewer #6: Yes

6. Review Comments to the Author

Reviewer #3: No corrections were made to the statistical analyses in Table 3, although the author acknowledged the error. Please reanalyze the data for categorical independent variables (gender, educational level, working shift) in relation to work engagement using t-tests or ANOVA, as appropriate, and present the revised results in Table 3. Currently, the categorical variables are still reported in correlation form, which is not suitable.

Reviewer #4: Overall, the manuscript addresses an important topic and is technically sound. With minor revisions to clarify statistical details and improve language flow, it has the potential to make a valuable contribution.

Reviewer #6: The title: good, recent study design

The abstract: clear and to the point

Methodology: Written in a good sequence and illustrative manner

The results: tables are well-organized and illustrative highlighting the main findings in a professional manner

Discussion: can be condensed to emphasize the study's recommendations but to some extent exhibited a satisfactory interpretation of the results

Limitation: the selection bias is not the only type of bias that may have been presented within this type of sample for example: the self-selection, the confirmation or the non-response bias. Denote each type by its name as they are already stated.

The conclusion: written in a professional manner

References: clearly written and recent

7. PLOS authors have the option to publish the peer review history of their article (what does this mean? ). If published, this will include your full peer review and any attached files.

**Do you want your identity to be public for this peer review?** For information about this choice, including consent withdrawal, please see our Privacy Policy .

Reviewer #3: No

Reviewer #4: **Yes: ** Mohammed Musaed Al-Jabri

Reviewer #6: **Yes: ** Dr. Lareen Magdi El-Sayed Abo-Seif

---

## [Author Response · Author response to Decision Letter 3]

4 Sep 2025

Thank you for your consideration and valuable feedback on our manuscript.

Please see the attached Response to Reviewers file.

We have provided detailed responses to each specific reviewer and editor comment in the attached document.

---

## [Editor Report · Decision Letter 3]

8 Sep 2025

Work Engagement and Its Association with Emotional intelligence and Demographic Characteristics among Nurses in Palestinian Neonatal Intensive Care Units

PONE-D-25-23550R3

Dear Dr. Ibrahim Aqtam,

We’re pleased to inform you that your manuscript has been judged scientifically suitable for publication and will be formally accepted for publication once it meets all outstanding technical requirements.

Kind regards,

Osama Mohamed Elsayed Ramadan

Academic Editor

PLOS ONE
---

## [Editor Report · Acceptance letter]

PONE-D-25-23550R3

PLOS ONE

Dear Dr. Aqtam,

I'm pleased to inform you that your manuscript has been deemed suitable for publication in PLOS ONE. Congratulations! Your manuscript is now being handed over to our production team.

Kind regards,

on behalf of

Dr. Osama Mohamed Elsayed Ramadan

Academic Editor

PLOS ONE